# Collaboration in East Africa: A Contextual Definition

Mónica Fontana [1], Francesca Peverelli [2] and Mauro Giacomazzi [3,*]

1 Facultad de Educación, Universidad Complutense de Madrid, 28040 Madrid, Spain
2 Avsi Foundation, Kampala P.O. Box 6785, Uganda
3 Luigi Giussani Institute of Higher Education, Kampala P.O. Box 40390, Uganda
* Correspondence: g.mauro@lgihe.org

**Abstract:** Collaboration is a crucial skill for the improvement of educational outcomes in adolescents. Culture affects the way people collaborate to solve problems, share challenges, and make decisions. Yet, there are only a few studies conducted in the African context that investigate local understandings of the concept of collaboration. This article aimed at investigating the concept of collaboration in the East African context and how this resonates with the local culture. The approach used to develop this study is the qualitative comparative method. One of the most relevant results from these analyses is working together or staying together as frequently used definitions of collaboration. The local definition of collaboration underscores a dimension related to being with another person; for the East African population, collaboration can be described as a way of living and facing reality in a community while being accompanied by others.

**Keywords:** collaboration; skill structure; contextualisation; East Africa; qualitative comparative method; ethnographic study

## 1. Introduction

Collaboration is a crucial skill for the improvement of educational outcomes in adolescents [1]. Research findings show that systemic collaboration in schools improves academic experience and social-emotional and behavioural outcomes (de Jong et al., 2022; Hargreaves, 2019). Many challenges can be faced and solved if all individuals work together [2]; if schools are to be effectively supported and empowered to fulfil their function of providing quality education, stakeholders' collaboration within the education sector is pivotal [3].

This article aimed at investigating the concept of collaboration in the East African context and how this resonates with the local culture. Collaboration is a key characteristic of humanity and it is a crucial element of our cognitive development [4]: "we are social beings and in a very significant sense, teamwork is part of our nature" [5]. In this regard, it would be very difficult to talk about collaboration without referring to the root works of Piaget [6] and Vygotsky [7]. Even if these theories were developed several decades ago, they are still relevant reference frameworks that can be used to investigate the meaning of collaboration. Piaget provided theoretical elements to explain cognitive progress with a theory of the four stages of cognitive development. Furthermore, he introduced the concept of cognitive conflict, which he brought to light as a mandatory step that triggers the process of understanding and comprehension of something new. Every time cognitive conflicts occur within a group of people who are collaborating, a series of interactions happen that can advance the progress of knowledge to a further level. On the other hand, Vygotsky's work did not specifically focus on cognitive conflict but placed more emphasis on the value of social interactions [8]. In his theory, individual cognitive change is caused through the interactions between a collaborator and the learner. For Vygotsky [7], knowledge happens in the zone of proximal development, a term which refers to the difference between what a learner can actually learn without help and what he or she can achieve only through guidance and encouragement from a skilled partner [9]. Overall, Piaget (cognitive conflict)

and Vygotsky (zone of proximal development), agreed that everyone needs both the environment and social interactions to gain knowledge about the self and the world.

However, presently, do we all understand the same thing when we try to define and foster collaboration? The definition of collaboration can be elusive or perceived as theoretical. Due to its overuse in referring to multifaceted social practices, the term collaboration has become a generic term used to signify almost any type of interpersonal or interorganisational relationships [2,10]. Collaboration is frequently used as a broad term that is associated with a set of constructs in regard to teamwork interactions across various settings. The definitions and use of terms tend to overlap and sometimes appear to be used interchangeably (e.g., teamwork, cooperation, partnership, strategic alliance), making it difficult to clearly define collaboration (Griffiths et al., 2021 [2]). Collaboration refers to a core competency essential to navigating today's learning settings and to thriving in the workplace [11], and it is "difficult to describe and yet more difficult to produce" [5]. To understand the meaning and impact of collaboration on our daily lives, we could just look around us and then ask ourselves whether something could have been made by only one person [12]. Even if the example of everyday objects helps us to realise how widespread the use of collaboration is, it is challenging not only to establish the definition of collaboration but also to pursue its enhancement (how to teach it) and assessment (how to measure it) [12].

## 2. Literature Review

Several other authors contributed to building a structured definition of collaboration [5,8,13,14]. Some authors highlighted relevant aspects of collaboration and presented a definition [12,13], and a recent systematic review synthetised the conceptual frameworks supporting the different uses of collaboration [2]. What emerges from these studies is that, as social human beings, most of our problems are solved by coming to terms with other people [15]. Thus, the fundamental need to form relationships can include both primarily interpersonal ties and primarily instrumental ties [15], which is to say, relationships formed to achieve shared goals. This points to a fundamental aspect of the nature of collaboration [12]: it can be improved and developed but it is already part of our nature.

A systematic review initiated in the 1990s and updated in 2018, which focuses on the needs of the labour context, defined collaboration as: "A mutually beneficial and well-defined relationship entered into by two or more organizations to achieve common goals" [13].

As highlighted in the above citation, a key element in the definitions of this skill is the aim of collaboration: the achievement of a common goal [5]. In other words, a collaborative process takes place when two or more persons engage in a joint activity to achieve a shared goal [16]. The same aspect is reported in Griffiths et al. [2] and in the Easel Lab [17] where the general definition of collaboration in the different frameworks respond to the act of working together to achieve a common goal. Achieving a common goal is not spontaneous and mechanical—it is not a mere assembling of parts—but it is a process that calls for the personal engagements of the actors [13,18]. To solve a problem, the mutual engagement of participants in a coordinated effort is necessary [18]. Participation, coordination, and individual contributions are not sufficient for effective collaboration; a personal commitment to the goal is required.

In the last 20 years, 65 systematic reviews or meta-analyses have been published on the topic of collaboration in education or in school settings. Several of them (almost 40% of the papers) are concentrated between 2017 and 2022, showing a growing interest of researchers in reaching a clear definition of collaboration, including related terms and supportive factors, in order to develop a conceptual framework in different fields. In education, two systematic reviews are particularly relevant for the topic tackled in this study: the systematic review of de Jong, Meirink, and Admiraal [19] about school-based collaboration as a learning context for teachers, and the one published by Griffiths et al. [2], in which a conceptual framework of collaboration is presented.

Griffiths et al. [2] presented the most common constructs that underpin the process of collaboration: open communication, trust, mutual respect, shared goals, common understanding, shared responsibility, active participation, and shared decision making. The aim of collaboration is to engage in meaningful interaction to achieve a common goal or to make shared decisions. To reach this result, relationship skills are crucial in order to create an environment in which people involved in the collaborative process can comfortably and effectively express themselves (open communication) [20,21]. Being flexible and adaptable are also important characteristics that the members should display. Usually, this happens when the people who collaborate develop a sense of mutual trust and they are willing to sacrifice their own interest in order to prevent the other person from being negatively afflicted [10,22,23]. In building positive relationships, it is also important to show mutual respect by understanding and valuing the levels of knowledge, skills, and competencies of the others [2]. As it is often mentioned in the definitions of collaboration, once the team has developed mutual trust and open communication, they usually determine a set of objectives and goals that they aim to achieve. This is always after realising that such objectives and goals cannot be achieved outside a collaborative process [21,24].

Related to this, the members of the collaboration team should regard each other with respect by showing mutual esteem demonstrated through actions and communications [21]. A team may have members with various expertise and cultural backgrounds but the collaborative process should aim to form a common understanding of the challenges and the strategies required to achieve shared goals [25]. This allows each member to participate in the collaborative tasks with his or her own expertise and competences according to the principle of shared responsibility. This helps in building a clear understanding of the roles of each member and in enabling collective ownership for the expected outcome [26], in the spirit of an active and meaningful contribution and participation of each member [24]. When these steps in the process are undertaken and all members are encouraged to actively participate, the decision-making process is shared and owned by all members [27]. At last, the effective implementation of the decisions made by the team is the proof of a positive collaborative process [2].

In summary, collaboration is a process that can be analysed and optimised; it is the act of working together to pursue a common goal [5,16] and all members are responsible for the entire process to achieve the goal [8,28]. These interactions between people lead to the growth of shared understandings because the process of collaboration includes the construction, control, and redefinition of one's knowledge [8,28].

This pursuit is not completely effective without both a genuine commitment of all people involved and a proper set of skills and dispositions [12,18]. The social domain of collaboration includes the skills of participation, perspective taking, and social regulation [12,29]. Participation mainly refers to a person's ability to positively interact with others; it requires the willingness of this person to share thoughts and information. Perspective taking—the ability to put oneself in somebody else's perspective [29]—facilitates the coordination process between collaborators and the appropriate responses to the various contributions. Social regulation refers to elements of navigating through differences and bridging the different perspectives.

As stated above, collaboration draws upon the knowledge, skills, values, and competences of the various members who contribute with their effort to the achievement of the common goal [13]. Yet, if we consider the set of skills and dispositions required for an effective collaboration, it should be underscored that members have different understandings and interpretations of how the collaboration process should take place and thus their experiences might be different [30]. This aspect becomes even more relevant when the participants in the process have different cultural backgrounds; culture influences the way in which people interpret and make meaning of the world and it affects the way people solve problems, share challenges, and make decisions [30,31]. In this regard, there are only a few studies conducted in the African context that investigate local understandings of the concept of collaboration [5,32]. Similarly, assessment tools and teaching techniques of this

skill are not sufficiently contextualised. A recent systematic review of the literature [33] underscored the need for studies that more accurately consider the higher cognitive effort that is needed in the collaboration process as compared to the individual learning process; its effectiveness depends on whether the result is worth the extra effort, which is needed in the collaboration process [33].

While the cognitive elements are specific to the context [31], they have not been investigated in the sub-Saharan continent where the understanding of the definition of these skills is still inadequate [32]. Moreover, in this context, there is a lack of assessment literacy among teachers and other stakeholders that is considered to be necessary for the development of standardised assessment tools for these skills [32]. Investigating an operational definition of collaboration in East Africa is thus an important step towards developing contextualised assessment tools and materials to support educators in the process of enhancing this skill. To bridge this gap, the aim of this study was to describe the conceptualisation of collaboration as a skill in the East African context, with the goal of identifying key elements of collaboration in this context.

### 3. The Context of This Study

This study is part of a broader project started by the Regional Education Learning Initiative (RELI) in East Africa in 2018. The project aimed to work with local leaders to collaboratively co-create and develop contextualised assessments in Kenya, Tanzania, and Uganda on life skills and values in East Africa. The initiative launched the Assessment of Lifeskills and Values in East-Africa (ALiVE) project in 2020 with the following goals: developing contextualised tools for assessment of life skills and values in East Africa and generating large-scale evidence to inform change and build capacity across the region.

The ALiVE project targets adolescents aged 13–17 (both girls and boys; both in and out of school), and it aims at assessing three skills—self-awareness, problem solving, and collaboration—and one value, respect. The purpose of this study was to obtain a contextualised understanding of collaboration skills in Kenya, Tanzania, and Uganda in order to determine the best tool for a large-scale evaluation of collaboration in the three countries. Therefore, this study presents a comparative analysis of the results obtained from the individual country's studies in order to identify the commonalities and unique features across the three countries.

### 4. Research Questions

The study aimed at answering the following questions:

(i) What are the key elements in the definition of collaboration as understood by people in Kenya, Tanzania, and Uganda?
(ii) What are the most common and unique subskills of collaboration highlighted among adolescents, parents, and key persons in the three countries?
(iii) What are the common and unique dispositions linked to collaboration as identified by adolescents, parents, and key persons in the three countries?
(iv) Which behaviours and values linked to collaboration are uniquely identified by adolescents, parents, and key persons in Kenya, Tanzania, and Uganda?

### 5. Methodology

The approach used to develop this study is the qualitative comparative method, which aims to establish similarities and differences across comparable cases [34] in the different dimensions and codes analysed. In regard to the study design, a qualitative approach and an ethnographic design was adopted in order to explore and collect participants' perceptions and understanding of collaboration in Kenya, Tanzania, and Uganda.

*5.1. Participants and Techniques of Data Collection*

The study was conducted in 5 districts in each of the three countries (15 total): Uganda (Jinja, Kikuube, Moroto, Kampala, and Oyam), Kenya (Rongo, Mwea East, Kibra, Narok

South, and Tana Delta), and Tanzania (Ilala, Mvomero, Ngorogoro, North-A, and Uyui). They were sampled on the basis of rural-urban economic activity (pastoralist, core-urban, and agricultural) and on the basis of distance from the respective capital cities.

In each district, two villages were randomly sampled. In each sampled village, at least 4 interviews with adolescents (2 of each gender, mixing those in primary and secondary school, vocational training, and those out of school), 4 interviews with parents (2 of these sampled adolescents and 2 of non-sampled adolescents, while mixing fathers and mothers), and 4 interviews with key persons (teachers, social workers, and others who consistently work with adolescents, mixing gender) were targeted. This resulted in a target of 24 participants for the one-on-one interviews per district. Overall, a total target sample was about 120 participants in each country for the interviews. However, given the prevailing challenges, the study sample for the one-on-one interviews varied: 116 participants in Kenya, 132 participants in Tanzania, and 120 participants in Uganda. Notably, not all participants were interviewed about collaboration. Only 75 participants in Kenya, 67 in Tanzania, and 95 in Uganda were interviewed about collaboration.

In addition to the one-on-one interviews, focus group discussions (FGDs) were conducted: 21 FGDs (10 for adolescents and 11 for parents) in Kenya; 20 FGDs (10 for adolescents and 10 for parents) in Tanzania; and 20 FGDs (10 for adolescents and 10 for parents) in Uganda. To constitute the FGDs, in each village, 3 participants (adolescents or parents) were selected to join the other 4 who participated in the interviews. Ultimately, FGDs in each village comprised of 5–7 participants.

### 5.2. Instruments of Data Collection

In-depth interviews: One-on-one interviews with adolescents, parents, and key persons were conducted to generate their understanding of problem-solving skills in each country's cultural context. The interviewers used an interview guide that was developed prior to data collection (see Appendix A).

Focus group discussions: Discussions with adolescents and parents were conducted to generate a deeper understanding of the issues that emerged from the interviews. Researchers used an FGD guide that was specific to a particular site and its interviews, and which was developed before data collection (see Appendix B).

### 5.3. Qualitative Data Analysis

A coding system was established in order to analyse the interviews and FGDs on collaboration following thematic analysis. Thematic analysis is "a method for identifying, analysing and reporting patterns (themes) within data" [35].

For the analysis of the interviews, we established a coding system based on contextual (descriptive) variables, including (1) the category of informants, (2) sex of the participants, (3) country, and (4) district. In quantitative terms, the contextual variables were analysed descriptively (frequency and percentage) using EXCEL and Dedoose.

The coding system also took into account thematic variables related to (5) the definition and process described by the participants, (6) subskills, (7) dispositions and values, (8) behaviours, and (9) support systems and factors for enhancing collaboration skills in adolescents. In qualitative terms, as recommended by Gibbs [36] and using the Dedoose program (version.8.3.41.), we performed an analysis of the understanding of collaboration as presented in the interviews.

These categories emerged from the analysis of five interviews (at least one from each category) conducted by nine research assistants in order to achieve the inter-rater reliability of the coding system. Apart from these predetermined categories, others emerged from the main topic of collaboration, and a unique network in understanding the skill was addressed in this report. The process of analysis involved the identification of patterns of similar ideas, concepts, or topics in order to establish the connection and integration of information with the theoretical foundation [37], as well as a suggested indication of or evidence for

contextualisation. The codes were created in accordance with the criteria for qualitative evaluation: dependency, transferability, credibility, and verifiability [38].

Furthermore, the analysis of the data collected in each country followed the three stages pointed out by Thomas and Harden [39]: the free line-by-line coding of the primary interviews, including sentences or paragraphs as the analysis unit; the organisation of these "free codes" into related areas to construct "descriptive" themes; and the development of "analytical" themes (p. 4). The analytical themes were then related to the recommendations for assessment, intervention, and policy making in order to contextualise collaboration skills in East Africa.

In addition, the triangulation technique [40] was carried out among the three researchers who analysed the data in the three countries in order to search, identify, select, evaluate, and summarise data from interviews based on predefined criteria and emergent categories. Moreover, the FGDs were used as a triangulation method for the data collected in the interviews conducted with the participants.

At last, data reduction was applied through mixed method analysis: (1) the initial subgroup classification of the interviews is based on the category of the participants adolescents, parents, and key persons); sex of the participants; and the districts; and (2) data reduction involves techniques of extracting and coding data. These mixed method analyses were carried out thanks to the Dedoose program, which facilitates analysis of the frequency of the codes in terms of the demographic information of the participants. In this regard, three types of descriptive analysis were carried out: co-occurrence of the codes, Descriptor x Code and Code x Descriptor, and Descriptor Field x Code Grid Chart.

### 5.4. Ethical Considerations

The research team utilised approaches that address ethical considerations in dealing with different categories of participants. These included: obtaining informed consent; ensuring confidentiality of information obtained from the participants; compensating (both monetarily and non-monetarily, i.e., offering readers to adolescents) participants; and ensuring voluntary participation, among others.

## 6. Results

To answer the research questions, the findings are presented according to the identified content variables related to (1) the definition and process described by the participants, (2) subskills, (3) dispositions, and (4) behaviours and values.

In reporting the findings, the excerpts from the dialogues with teachers are reported with reference to a code that is composed by the first letter identifying the country the participant is from (e.g., K = Kenya), the second letter identifying the category of the participant (i.e., K = key persons, P = parents, A = adolescents), and the number of the interview.

### 6.1. Key Element in the Definition of Collaboration as a Life Skill

The following Table 1 is related to the codes and the analysis of the definitions of collaboration as given by the participants in the three countries and shows the most meaningful codes in developing a common definition of collaboration and also the specificity for each country. The selected codes have been mentioned by at least 10% of the participants.

The comparison among the codes that appeared in the reports related to Uganda, Kenya, and Tanzania highlights some similarities and differences that deserve to be analysed. When the participants defined collaboration in the three countries, the most common code that emerged from the interviews is working or staying together. This code is very broad because it covers both the dimension of work and that of being together. It is a code that recalls community life in all of its features, which are both those of living or being together and those of working. This explains why this code has been distinguished from teamwork or cooperation, which is more related to work or school settings.

**Table 1.** Frequency of the definition's codes mentioned by at least 10% of the participants.

| CODES | KENYA (Participants) | | TANZANIA (Participants) | | UGANDA (Participants) | |
|---|---|---|---|---|---|---|
| | **Freq.** | **%** | **Freq.** | **%** | **Freq.** | **%** |
| Working/staying together | 59 | 78.66 | 64 | 71.64 | 64 | 77.89 |
| Teamwork/cooperation | 11 | 14.66 | 22 | 16.41 | 22 | 23.15 |
| Helping the community | 11 | 13.33 | 22 | 32.84 | 10 | 10.52 |
| Sharing | | | | | 18 | 18.94 |
| Goal setting | 10 | 13.33 | | | | |
| Relationship skills | 10 | 13.33 | | | | |
| Unity | 9 | 12 | 21 | 31.34 | | |
| Agreement | 9 | 12 | | | | |
| TOTAL OF PARTICIPANTS | 75 | | 67 | | 95 | |

Note: Since each participant mentioned more than one code in their definition of collaboration, the percentage does not sum up to 100. This number has been calculated based on the total number of participants in each country.

Several participants from the three countries highlighted in their responses that collaboration is togetherness, while a smaller number (only 10 participants) mentioned that it means to achieve a common goal. In fact, according to the participants, the achievement of a common goal is not always the motivation for working together.

A key element that emerged from the analysis of collaboration codes concerns the link between skills and values. In defining collaboration, the participants refer to sharing, unity, and love. Even though collaboration is defined as working together, the value dimension plays an important role. Belonging to the community is a key element of collaboration; participants themselves feel part of the community. The term unity emerged as evident from the interviews in Tanzania. Unity is understood as a term synonymous with collaboration: being collaborative and moving in unity are synonymous according to 31% of the interviewees. This occurrence emphasises again how the concept of collaboration is strongly communitarian and is a recurrent theme in the process of defining this skill.

In regard to sharing, a number of interviewees (18%) highlighted this aspect that pertains to their daily life. It is not just about sharing material things in case of need but it covers the whole sphere of trustful interaction among peers and mutual listening and understanding. It is not necessarily aimed at solving problems but it is a form of help and peer support as a way of living, as to be in groups, sharing (U-K-35). This response reveals that being in a group is a great situation to share problems. The aim of this sharing is often to receive advice from other experienced community members. Similarly, collaboration takes place through sharing because "it has to do with convening people with a mission to have a pressing need discussed and resolved" (U-K-37). Collaboration here means conveying decisions that have consequences for others.

Relatedly, a relevant element concerning the definition of collaboration is helping community. In all three countries, the excerpts mentioning this are above 10% and, in Tanzania in particular, they exceed 30%. This is because helping and supporting the needs of the community to which one belongs is a very strong motivation to collaborate. In fact, it emerges from the concrete examples offered by the participants that community members are concerned about each other's needs: "Concerning problems, it is the work of parents to come together and solve the issue. For example, when there is flooding, people come together and build a dyke to prevent the village flooding" (K-K-18).

For this reason, some participants identify collaboration as finding solutions; solving concrete problems is often the element that triggers collaboration among community members.

The high number of cross-cutting occurrences of helping community and teamwork codes confirms this communitarian conception of collaboration. Teamwork is also a cross-cutting code that appears with similar occurrences in the three countries, though it is more

present in the interviews from Uganda. Several times, when participants were asked to state any other word that means the same as collaboration, the answer given was teamwork or cooperation.

Lastly, goal setting and relationship skills have a few occurrences in Kenya (13%) as compared to none in Tanzania and Uganda. Relationship skills are deeply connected to collaboration because collaboration is a skill that occurs in continuous relationships with others.

### 6.2. Subskills of Collaboration

In the context of our study, subskills are understood as the skills that are part of a more complex skill, in this case, collaboration. The following table shows the list of similarities and differences among the three countries of the codes identified as collaboration subskills.

The classification of subskills that emerged from the codes identified during the analysis of the documents in the three countries is presented in Table 2. This process reveals similarities and differences that highlight some interesting aspects of the contextualised understanding of collaboration. Teamwork is identified not only in the definition of collaboration but as a subskill of collaboration numerous times in Uganda and Kenya (the percentages are close to 50%); however, it does not emerge in Tanzania.

**Table 2.** Frequency of the subskills' codes mentioned by at least 10% of the participants.

| CODES | KENYA (Participants) | | TANZANIA (Participants) | | UGANDA (Participants) | |
|---|---|---|---|---|---|---|
| | Freq. | % | Freq. | % | Freq. | % |
| Relationship skills | 28 | 37.33 | 12 | 17.91 | 49 | 51.57 |
| Receptive communication | 20 | 26.66 | 9 | 13.43 | 30 | 31.57 |
| Guidance & counselling | 16 | 21.33 | 19 | 28.35 | 21 | 22.10 |
| Teamwork/cooperation | 35 | 46.66 | | | 48 | 50.52 |
| Expressive communication | 17 | 22.66 | | | 29 | 30.52 |
| Goal setting | 15 | 20 | 7 | 10.44 | | |
| Self-confidence | 10 | 13.33 | | | | |
| TOTAL OF PARTICIPANTS | 75 | | 67 | | 95 | |

On the other hand, relationship skills, communication, and guidance and counselling are identified as cross-cutting subskills in the three countries. The ability to relate adequately—to be expressive and receptive in terms of communication—are abilities identified as relevant subskills to collaborate. In fact, collaborating is something that concerns others or, in other words, the relationship with other persons.

Relationship skills appear in all three countries as characteristics that identify an effective collaborator: collaborating means "making an alliance with others; associating and cooperating" (U-K-40) and "collaborating is having good relationships" (T-A-33). Similarly, in Kenya, an adolescent explained that collaboration is to take care of relationships and try not to give rise to misunderstandings among peers (K-A-18). Being social is highlighted as one of the most typical aspects of the collaborator. In this sense, the characteristics of an effective collaborator are frequently phrased as "he should be a very social person who loves doing things with other people, but not alone, and one who likes sharing with everyone regardless of religion" (U-A-07). Similarly, collaboration is identified as a trigger to socialisation, which is the first step to collaboration (T-K-19). It is relevant to note that there is a theme of overcoming social and cultural barriers to collaborate. Similarly, the behaviour expected of an effective collaborator is being social and interactive toward people: "they are interactive even with new people" (U-A-02). A collaborator is described by some

adolescents as someone who is naturally inclined to pay attention to the needs of the other, for instance, someone who "has to have the desire of being helpful to others" (U-A-11).

Various participants reported a link between collaboration and relationship skills, adding the element of respect as a key element in the relationship. This link between skills and the value of respect highlights reciprocal esteem as the basis for a collaborative relationship. In fact, the exercise of skills is not enough; collaboration is not a matter of merely executing a task but it fosters unity among the members of the community. For instance, "this person should be inspiring, able to interact freely and this person should have respect for himself and others" (U-K-17); participants from Tanzania state that where there is respect, there is caring for others, and this can be the beginning of a positive relationship (T-K-05 and T-P-12). An adolescent in Kenya stresses the same issue, saying that an effective collaborator is pictured as someone who "shares with me his experiences during life skills sessions. Sharing and caring, you have to share your problems so as to get help. There are those you can share with and they despise you but if its him he urges one to not give up and encourages you to pray more" (K-A-29).

Communication helps to define the characteristics of an effective collaborator who must be attentive and approachable through dialogue. Effective collaboration always takes place through communication that is able to meet the other person in both listening and responding (U-K-14). Similarly, interviews from Kenya emphasise that a person with collaborative skills is someone who likes to talk to others and listen to others (K-A-01, K-K-07, and K-P-38).

This link between collaboration and communication also clearly emerges in the interviews from Tanzania where collaboration is, in a sense, having the ability to communicate. A participant from the key persons category shares an example that sheds light on the relationship between collaboration and communication. He argues that in order to help a child collaborate, one must first talk to him or her. The participant says: "You talk to him/her because there are others having some problems, so the first strategy is to talk to him or her, you can talk to him/her because others may not be aware of the importance of collaboration" (T-K-13).

Being welcomed in a group plays a key role in triggering the collaborative process. Although they used slightly different terms (listening skills, being an effective listener), many participants insisted on receptive communication or, rather, on the ability and willingness to listen. This spontaneous connection between listening skills and collaboration highlights the awareness of the fact that there is need for these skills to contribute to a common mission.

Listening enriches the self with the experience of the other. This awareness that grows in listening to each other means that the experience of each member of the community also enriches the others. The process described goes in both directions: requesting counselling and offering advice.

According to some adolescents (U-A-08, U-A-35, U-K-12, U-K-12, T-A-31, T-A-14, and K-A-03), it is precisely the fact of giving advice that characterises a peer as an effective collaborator. The collaborator is the one who is capable of offering his or her contribution to peers.

An excellent practical example of this flow of asking and giving advice within the community is presented in the following excerpt:

> To show that I am collaborative, let's say I am lacking something I should be able to go to my neighbour and request for it, and they give it to me. In case my fellow is having problems in her home and she wants to quit her marriage, I should be in position to advise not to leave and help her settle the issue. (U-K-13)

*6.3. Dispositions*

The theme of dispositions presents aspects of a person's character that help in enhancing and nurturing the exercise of collaboration skills. In the following tables, a comparison of the frequency and percentage of dispositions and values in the three countries is pre-

sented. It is important to note that some of the dispositions can also be treated as subskills. However, in this case, we are adhering to the participants' responses mentioned during the interviews (see Table 3).

**Table 3.** Frequency of the dispositions' codes mentioned by at least 10% of the participants.

| CODES | KENYA (Participants) | | TANZANIA (Participants) | | UGANDA (Participants) | |
|---|---|---|---|---|---|---|
| | Freq. | % | Freq. | % | Freq. | % |
| Hardworking | 9 | 12.00 | 13 | 19.40 | 21 | 22.10 |
| Leadership | 19 | 25.33 | 18 | 26.8 | 14 | 14.73 |
| Kindness/friendly | 17 | 22.66 | | | 26 | 27.36 |
| Willingness to be advised/corrected | | | | | 13 | 13.68 |
| Positive attitude | 10 | 13.33 | | | 13 | 13.68 |
| Responsibility | 14 | 18.66 | | | | |
| TOTAL OF PARTICIPANTS | 75 | | 67 | | 95 | |

Being hardworking and having leadership skills are dispositions that appear uniformly in all three countries where the study was conducted. Positive attitude has a significant number of excerpts in Kenya and Uganda but not in Tanzania. Similarly, being kind and friendly has a relevant number of excerpts in the same countries. Leadership and positive attitude are dispositions oriented towards others. Being hardworking, on the other hand, is a disposition of the self, independent of the relationship with others.

Being a hardworking person is a characteristic of a collaborator according to participants belonging to all categories (parents, key people, adolescents). It is interesting to note that being hardworking means having an active presence at various levels, from study to domestic help to supporting community needs. To exemplify the attitude of a collaborative teenager, a parent says: "She always used to help her mother with domestic activities and also doing business, emphasising her young brothers and sisters to work hard instead of depending . . . " (T-P-12).

Furthermore, in Tanzania, being hardworking has a connotation linked to moral conduct (T-A-15, T-A-12, T-P-14, T-P-1, T-P-08, T-K-08, and T-A-12), which does not appear in the interviews from the other two countries. In fact, it is explained as being "dedicated to work" (T-A-15) and it is also linked to the values of respect for adults (T-P-14), discipline, and obedience (T-P-08).

According to the adult participants, "when you give them [youth with collaborative skills] an activity, they don't waste time, they take the activity very fast and positively. The child also has more ideas; he does not need to be told everything" (K-K-35). In a similar way, parents said that a collaborative youth "always works on jobs available in the community" (K-P-01). In Uganda, it should be noted that six of the participants who identified being hardworking as an important disposition to be an effective collaborator are teenagers. When asked about the evidence of collaboration of a peer, they said that he or she is collaborative precisely because he or she is hardworking. In other words, being collaborative is typical of someone who is hardworking. Similarly, some parents questioned about the behaviour of a collaborative teenager replied, "these adolescents have good listening skills. Does home chores and they are hardworking" (U-P-38).

Being kind or friendly and having a positive attitude have an important number of occurrences in Kenya and Uganda, highlighting how these personal characteristics favour the collaborative process as they spontaneously lead to meeting and interacting with others.

### 6.4. Behaviours and Values

The following Table 4 presents the frequency and percentage of the most common codes that emerged in the three countries. As can be observed, behaviours and values are combined in the same list and maintained as presented by the participants.

**Table 4.** Frequency and percentage of codes identified as behaviours and values of collaboration in Kenya, Uganda, and Tanzania.

| CODES | KENYA (Participants) | | TANZANIA (Participants) | | UGANDA (Participants) | |
|---|---|---|---|---|---|---|
| | Freq. | % | Freq. | % | Freq. | % |
| Positive behaviour | 59 | 78.66 | 57 | 85.07 | 60 | 78.94 |
| Respect | 23 | 30.66 | 17 | 25.37 | 26 | 34.21 |
| Love | 22 | 29.33 | 20 | 29.85 | 13 | 17.10 |
| Discipline | 10 | 13.33 | 12 | 17.91 | 14 | 18.42 |
| Obedience | 9 | 12 | 17 | 25.37 | 13 | 17.10 |
| Exemplary | 4 | 5.33 | | | 8 | 10.52 |
| Trust/honesty | 9 | 12 | | | | |
| Humility | 7 | 9.33 | | | | |
| Helping the community | | | 22 | 32.83 | | |
| TOTAL PARTICIPANTS | 75 | | 67 | | 95 | |

In the three countries in which the study was conducted, the cluster of behaviour and values is quite homogeneous. In fact, most of the codes are mentioned in all places by the various categories of participants. It is interesting to note how appropriate conduct cannot be separated from community values.

First, collaboration is associated with good behaviour by almost all participants. This is then followed by other values, such as respect, which contains percentages between 25 and 35, and then by love (17–20) and discipline (13–19).

The responses are generally descriptive of practical attitudes and collaboration is associated with a set of values that have a high worth of social cohesion. The definition of good behaviour shared by most of the participants included, "Good behaviour traits like helping others and sharing things like food. . . . One needs to have good discipline and be willing to share things in order to be considered collaborative" (U-A-11) or phrases using different words saying that being collaborative means being ready to contribute and give (U-K-04, K-A-13, K-A-15, and K-A-18). Particularly interesting is the element of readiness and attentiveness to the needs of the community as a unique component in their understanding of collaboration. A collaborator is present and active to the members and the needs of the community. In Tanzania, the fundamental aspect of collaborative behaviour is obedience and there are various examples that provide concrete situations in which the help of young people must be coordinated by a parent:

> Collaborative children in the community do things exactly to what they are directed to do. For example, when you are doing an event and you ask a child to do some chores, then they do without changing, we can now say they are cooperating with us. (T-P-25)

Respect, together with love, discipline, obedience, and being exemplary, are values recognised as fundamental according to the participants of various categories in all three countries.

## 7. Discussion

In relation to the first research question, one of the most relevant results from these analyses is working together as a frequently used definition of collaboration, with no emphasis on the final accomplishment. Working together is also the most common phrase used in the definitions of collaboration in the published articles of the last ten years [2], yet it is a very generic term. On the other hand, the local definition of collaboration as staying together underscores a dimension related to being with another person. In other words, for the East African population, collaboration can be described as a way of living and facing reality while not being alone, in a community accompanied by others.

With regard to the second research question, pertaining the subskills of collaboration, for the three countries, being together is inspired by need and not by a goal. The literature underscores that at the centre of the collaborative process, there is a need to achieve a shared goal or to solve a common problem; the team members are brought together to complete a task that could not otherwise be accomplished independently [8,12–14,32,41]. Interdependence is commonly established by capitalising on each other's contribution and when the overall tasks of the team cannot be completed without everyone working together [42]. Yet, the participants in this study describe collaboration more as the result of a concept of self as naturally bound to the other person more than an interdependence born out of a need to achieve a common goal. Since a person belongs to a certain community, that person collaborates with the members of that community; the purpose is intrinsic, not extrinsic. It should be noted that the definitions that emerged in the interviews connect with the etymological meaning of collaboration. Collaboration comes from a Latin root *cum* and *laborare*, meaning to work together. Similarly, Ofstedal and Dahlberg [43] assert that "people who practice true collaboration create a shared vision with joint strategies when working on a problem, issue or goal" (p. 38). Therefore, in the East African context, it is important to underline that collaboration is not only practiced because of the existence of a common goal, but in many cases, it is understood as the common way of living together while sharing experiences or helping in the community; it is a way of conceiving the self in relationship with the community, as also stated by Kim and Care [32]. Probably one of the most relevant findings about the contextualised understanding of collaboration in Kenya, Tanzania, and Uganda is the emphasis on the sense of community [44] or sense of belonging that is found in most of the definitions, explanations, and characterisation of a collaborative person. This aspect resonates with the main attributes of the *Ubuntu* philosophy, which promotes solidarity over individual self-sufficiency. According to this philosophy, a person matures while interacting with the other, through others. For this reason, the wellbeing of the community is considered even more important than the wellbeing of the single person since it is the community that, in the long run, ensures the wellbeing of each member. People are closely interconnected with the community and an individual is expected to learn through imitation and observation more than questioning [45,46]. The communitarian vision of life is somehow in opposition to the western vision of life which is more individualistic. This influences the way people face challenges and solve conflicts. In western societies, people tend to face conflicts in a more rhetorical way while African cultures promote approaches that avoid conflicts and discourage argumentation. Thus, collaboration is not reduced to accomplishing tasks in the school or work settings but is seen as a way of living and conceiving oneself. This awareness was reflected in the interviews through use of the following expressions: working with others, togetherness, helping community, sharing, unity, etc.

The findings on the third research question show that collaboration involves particular dispositions, such as leadership, positive attitude, and responsibility among others. These personal factors—though key to the collaboration process [19]—are important in the model of collaboration as it emerged in this study specifically through how individual competences can better enhance the good of the whole community.

In relation to the last research question, even if there was no specific question about values, the participants in the three countries mentioned respect, trust, love, and unity

as prerequisites for collaborating with others. As highlighted in the literature, trust is a key element in the process of building positive relationships among the members of a community—together with open communication and mutual respect—as well as common understanding [2]. In Kenya, the participants also highlighted the aspect of finding an agreement among members, which is linked to the aspect of finding a common understanding.

The comparative analysis carried out in this study highlighted some peculiarities typical of each country. Tanzania stands out for the importance given to unity in the definition of collaboration. It is no coincidence that it emerges precisely when the participants are asked for a synonym for collaboration. Unity therefore falls within the very essence of this skill. In support systems, it is interesting to note that the participants from Tanzania highlighted the most operational and practical aspects of collaboration, which are helping the community and sharing, as if it were the exercise of collaboration that enhances unity. Kenya, on the other hand, is the only country in which many dispositions are associated with being respectful, courageous, responsible, and patient, as well as with self-actualisation. The literature highlights the importance of shared responsibility in the process, underscoring the unique role that each member plays in defining potential solutions [2]. This implies a sense of collective ownership for the outcome and a clear sense of the roles and responsibilities each member has to play [27]. Lastly, in Uganda, it is interesting to note how relationship skills as part of collaboration subskills were discussed as fundamental for more than 50% of the participants. The relationship skills take many forms, including expressive communication and listening to others. However, it is interesting to see how these are necessary to collaborate with a willingness to encounter while remaining open to others.

According to the literature, the final stage of collaboration includes shared decision making and the effective implementation of these decisions [2]. Nevertheless, "although a crucial aspect of the team process, this concept in collaboration often did not appear in the literature. In fact, although briefly mentioned a few times, it was not a focus of any of the articles that we reviewed" [2]. Similarly, even the aspect of applying short- and long-term implementation activities [47] to facilitate the achievement of the goal is not mentioned by the participants of this study. These last steps that conclude the collaboration process are missing from the East African conceptual framework.

## 8. Conclusions

The limited amount of available empirical research on collaboration in East Africa, and the inconsistencies in terminology used across these studies, called for a study that was grounded in the experience of teachers, youths, educators, and local authorities in East Africa. For this reason, the objective of this study was to investigate how Kenyans, Tanzanians, and Ugandans understand and conceptualise collaboration. The culture, tradition, and the way of collaborating at the local level provides specific features on the perception of this skill. Without a strong and contextualised operational definition of collaboration as a skill, it is challenging for researchers to compare results across studies or contexts and distil best practices that are capable of contributing to the advancement of this field of research.

This study offers a synthetic contextualised elaboration (or deconstruction) of collaboration as a skill by presenting a structure of contributing subskills and dispositions that can facilitate the process of designing assessment tasks [29]. This skill structure can also be used in the process of identification of lower to higher competency in order to facilitate the nurturing of collaboration skills in schools and communities [12] and, ultimately, contribute to the improvement of students' learning outcomes. It will be important to evaluate how the elements of the collaboration skill structure interact with one another and with collaboration as a whole and how they are correlated to students' outcomes.

Moreover, these findings capture the unique understanding of the concept of collaboration in the East African context. A contextualised approach to investigating this skill is not only relevant from the anthropological and cognitive developmental point of view [30,31]

but it also builds the foundations for the incorporation of innovative approaches to the measurement and enhancement of life skills in the East African education systems. For instance, in the classroom, teachers could use the findings of the study as a basis to build a taxonomy for a formative assessment of collaboration or to develop a questionnaire to help in the development of collaboration skills among students. This instrument can be designed as a criterion-referenced list of attributes that are key to improve the culture of collaboration among team members, which is necessary in order to build a collaborative culture in a team. Each member may use the checklist and, as a group, the members may review the results and plan for strategies to improve their ability to collaborate [2].

This study also suggests that policies are more likely to be effective if accompanied by pedagogical support to teachers for their continuing professional development based on their own or context-specific understanding of the skills and based on more suitable methodologies for nurturing them.

**Author Contributions:** Conceptualization, M.F. and M.G.; methodology, M.F.; validation, M.F., F.P. and M.G.; formal analysis, M.F. and F.P.; investigation, M.F. and M.G.; resources, M.G.; data curation, M.F.; writing—original draft preparation, M.F., F.P. and M.G.; writing—review and editing, M.F., F.P. and M.G.; supervision, M.F.; project administration, M.G.; funding acquisition, M.G. All authors have read and agreed to the published version of the manuscript.

**Funding:** This research was funded by WELLSPRING PHILANTHROPIC FUND: 14849, ECHIDNA GIVING: 07.31.20, IMAGINABLE FEATURES: 03.25.20.

**Institutional Review Board Statement:** The research protocol was reviewed and approved by Mildmay Uganda Research Ethics Committee (Registration number: MUREC-2022134). The research team also secured approval from the Ministry of Education and Sports (MoES), a regulatory body whose mandate is to provide technical support, guidance, coordination, regulation, and promotion of quality education, training, and sports in Uganda.

**Informed Consent Statement:** Written informed consent was obtained from all subjects involved in the study.

**Data Availability Statement:** Data supporting reported results are available on request.

**Acknowledgments:** We acknowledge the intellectual contribution of John Mugo (Zizi Afrique Foundation), Purity Ngina (Zizi Afrique Foundation), and Martin Ariapa (Luigi Giussani Institute of Higher Education).

**Conflicts of Interest:** The authors declare no conflict of interest.

## Appendix A

*Interview Guide*

C.  Definition of collaboration
C1  Explain the word "collaboration".
C2  State other words that mean the same with collaboration.
C3  How is collaboration called in your local language?
C4  In what ways can you become more collaborative?
D.  Think of an adolescent (that is, a teenager of about 13–17 years old) that you know who has strong collaboration skills. He/she does not have to have a perfect life, he/she just has a clear and strong collaboration skills. You don't have to tell me their name. Take a moment to think about that person.
D1  Is this adolescent male or female?
D2  How old is this adolescent?
D3  Why do you think he/she has strong collaboration skills? How do you know, that this person has collaboration skills? For example, can you tell me what he/she has done?
D4  From what you mentioned, what do you think is the most important sign (indicator) that he/she has strong collaboration skills?
D5  What can you do, to test if a young person has collaboration skills?

D6 (a) Which behaviours are expected of a young person (aged between 13–17 years) who is collaborative? What do they do? (b) How about a person of same age, who has no collaboration skills?

D7 Which of the support systems (such as school, family, community, peers, friendships, etc.) do you think help this adolescent to have strong collaboration skills?

D8 What skills or attitudes do you think are important for an adolescent to be collaborative?

**Appendix B**

*Example of FGD Guide*

1. Sometimes one can solve a problem by themselves and other times by collaborating with others. What do you gain from collaborating with others in solving a problem? Explore.

2. What skills or attitudes do you think are important for an adolescent to be collaborative? Probe as much as possible.

3. What kind of problems does this adolescent normally face in his/her daily life (e.g., at home, school, community, with friends, etc.)?

4. What steps does he/she normally follow in solving these problems?

5. Which support systems (such as school, family, community, peers, friendships, etc.) do you think help this adolescent to solve problems? Probe: How?

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
