# Peer review of "Collaboration in East Africa: A Contextual Definition"

_education, doi:10.3390/educsci12100706_

Round 1

Reviewer 1 Report

A very interesting article! The topic is timely, the article is logical and written in good English. That said, I have a few remarks that might improve its overall quality. 

1. I don't know why there is no disctinction between INTRODUCTION and LITERATURE REVIEW. That makes the section too long. 

2. There are as many as 6 research questions (RQs). RQ 1 and RQ 2 can overlap, as there are no precise definitions what 'collaboration' means and what 'research skills' means. The answers to RQs are given in separate sections (which is right) but this only refers to subskills, dispositions and behaviours and values. It would have been clearer if the sections related to particular RQs (one RQ to one section), which might have involved the reformulation of RQs and the section part. The same notions of 'unique facets' and 'unique features' could have been used in the findings to better direct the reader to a particular question. 

3. How about attaching appendices with 1) Examples of questions that were asked during the interviews (at least on collaboration) and 2) One related to the FGD? 

4. How were participants compensated 'non-monetarily'? 

5. In the Discussion section I suggest an explicit reference to RQs. 

Author Response

Thanks for your kind suggestions.

The responses are in the attached file.

Cordially 

Reviewer 2 Report

In this manuscript, the authors set to investigate how East Africans (Kenyans, Tanzanians, and Ugandans) understand and conceptualize “collaboration”. The culture, tradition, and the way of collaborating at local level give specific features on the perception of this skill. They go about achieving this aim through conducting qualitative research with one on one interviews and focus groups, in each of the three countries. With their research, some themes that directly relate to the concept of collaboration start emerging. Each of those themes is then well-discussed and viewed within the context of East African cultures. 

Overall, this manuscript is well-written. The introduction section is comprehensive and prepares the reader to what is about to be presented. The used references are adequate, although many of them are rather dated. The presentation of the methodology, the results and their discussion are all well described and analyzed. The authors do in fact succeed at offering a synthetic contextualized de-construction of collaboration as a skill by presenting a structure of contributing subskills and dispositions. This may make it easier to design collaboration assessment tasks in future studies.

My comments are as follows:

1. In line 328, I am not sure what the authors mean here when using the word “confrontation”. Is this the right work to use, or did the authors meant something else and that is simply a typing mistake.

2. Since the manuscript is related to East Africa and collaboration, I was expecting to find a heavy focus of and use of the word “Ubuntu” and its philosophy as “humanity towards others”. However, the authors do not related their work to Ubuntu at all.

Author Response

Thanks for your kind feedback

The responses are in the attached file

Cordially 
